# Toward Sustainable ICT-Supported Neighborhood Development—A Maturity Model

**Madeleine Renyi** [1,2,*] **, Anna Hegedüs** [3] **, Edith Maier** [4] **, Frank Teuteberg** [2] **and Christophe Kunze** [1]

1   Care & Technology Lab (IMTT), Furtwangen University, 78120 Furtwangen, Germany;
    christophe.kunze@hs-furtwangen.de
2   Department of Accounting and Information Systems, Institute for Information Management and Information
    Systems (IMU), Osnabrück University, 49069 Osnabrück, Germany; frank.teuteberg@uni-osnabrueck.de
3   Careum School of Health, 8006 Zürich, Switzerland; anna.hegedues@careum-hochschule.ch
4   Department of Economics, University of Applied Sciences of Eastern Switzerland,
    9001 St. Gallen, Switzerland; edith.maier@ost.ch
*   Correspondence: madeleine.renyi@hs-furtwangen.de

**Abstract:** Scientists promote the fostering of caring communities as a means of sustainably coping with demographic changes. They consider community-based technologies to have a high potential for supporting the establishment of caring communities. However, implementing community-based technologies is a complex endeavor, making sustainable adoption difficult. We have developed a maturity model aimed at standardizing the monitoring and evaluation of neighborhood projects. Based on a maturity model for integrated care, we conducted a Delphi study, to develop a maturity model for information and communication technology (ICT)-supported neighborhood development. In eight case studies, the model was validated and subsequently adapted to the specific needs and requirements of neighborhood projects. These studies emphasized the importance of at least 12 decisive dimensions and showed that the maturity model could be applied at different stages of a project. The current form of the maturity model can be used to help orient, as well as monitor and evaluate, neighborhood development projects. Future work will be necessary to further develop accompanying materials and services and to foster the exchange of best practices and experience between projects.

**Keywords:** maturity model; neighborhood development; ICT—information and communication technology

## 1. Introduction

Many people prefer to grow old at home in a familiar environment. However, changing social patterns, such as long-distance family ties or increased women's education and employment [1], represent major challenges to in-home care. Politicians and socially engaged groups are developing, discussing, and promoting a wide range of countermeasures, such as neighborhood projects intended to foster social sustainability [2]. In caring communities, each individual, as a member of his or her neighborhood, supports the local community through demonstrating co-responsibility, engaging in co-production, and recognizing interdependence and reciprocity [3]. These communities promote solidarity and a shift toward a more socially sustainable distribution of care tasks.

The popularity of the concept of caring communities and closely related approaches, such as compassionate communities and cities [4], reflects an increasing focus on neighborhoods and communities as key sites for social inclusion and participation [5,6] and the importance of solidarity-based

relationships and networks for healthy living [7]. According to gerontological research, the quality of one's social networks influences life expectancy more than biological parameters [3]. An increasing number of municipalities are realizing not only that digitization promises to make processes more efficient, reduce media disruptions, and conserve resources but also that technologies for community collaboration have the potential to create a collective awareness of co-responsibility, co-production, and reciprocity and to increase the visibility and awareness of neighborhood resources. Particularly during the current Coronavirus pandemic, digitization, in the form of, for example, telemedicine and videoconferencing, can help reduce the number of personal contacts, while allowing people to remain in contact virtually.

However, practitioners and researchers have repeatedly indicated, how difficult it is to set up sustainable digitization projects in neighborhoods and evaluate their outcomes. Despite this difficulty, potential funding organizations and authorities often require that such projects be evaluated according to standardized procedures. Community structures, goals, and local conditions tend to differ and are therefore difficult to compare. Although many reports exist regarding neighborhood development and individual projects, there are few evaluations of the effectiveness of digitization projects. Despite a broad selection of evaluation tools from a wide range of disciplines (e.g., sociology, business informatics, health science), it is challenging to develop a suitable evaluation strategy for a technology-based community intervention. Even though randomized controlled trials (RCTs) are frequently regarded as the gold standard, many studies that focus on community-based interventions have reported methodological difficulties with RCTs and, in particular, have noted the difficulty of randomizing neighborhoods [8,9]. Therefore, in the context of neighborhood development interventions, it may be favorable to evaluate the prerequisites as well as the underlying structures and processes instead of the outcomes.

The objective of this article is to outline the development of a maturity model for information and communication technology (ICT)-supported neighborhood development. With the B3-Maturity Model (B3-MM) [10] serving as a starting point for its development, the underlying research questions are as follows:

*Is the B3-MM transferable to the neighborhood context? If so, which adaptations are necessary and which application scenarios are conceivable?*

Following this introduction, we discuss the broader background regarding digital collaboration tools for community development, the process of introducing such technology into neighborhoods and the difficulties involved in doing so (Section 2). Also, we briefly present the B3-MM, which was chosen as a starting point for the development of our maturity model. Section 3 describes the methods we used for developing and evaluating the maturity model, while Section 4 details the maturity model tailored to neighborhood development. Subsequently, we outline the application possibilities in terms of the model, target groups, and possible objects of investigation (Section 5). Finally, we summarize our results and draw conclusions (Section 6).

## 2. Background

### 2.1. Community Development

Community development is significantly influenced by how challenges are addressed [11]. To strengthen social cohesion in local communities, joint efforts are needed from all relevant actors [12,13]. There is a broad spectrum of fields of action in neighborhood. Whether rural or urban, the common goal is to develop a lively social community with strong civic involvement [3,14,15]. The following fields of action are commonly in the foreground when deriving suitable measures and priorities for sustainable neighborhood development: participation and engagement; care and support; housing, living environment and mobility; family and generations; local economy and professions; health promotion and preventative care; integration; and people with disabilities and inclusion [3,11–16].

Successful neighborhood development requires that people be involved in the process [13]. Due to demographic and social changes, older people move into the focus of action. Age-friendliness is a main objective [16]. With an increase in women's employment, social relationships and cohesion in families decrease [1]. The local economic area, which determines the quality of life and living, and the potential of each individual to help themselves, must be promoted. One must clarify that taking in people from different backgrounds is not only a challenge but also opportunity for the whole community. The coexistence of people within a neighborhood leads to enriching diversity.

*2.2. Digital Tools for Community Collaboration*

The term collaboration derives from Latin and means working together (*co* = together; *laborare* = to work). People who work together usually do so to increase the efficiency and effectiveness of their joint efforts to achieve their common goals [17]. In neighborhoods, various technologies can be used to achieve the abovementioned community development goals of co-responsibility, co-production, and reciprocity. Generally, these tools can be divided into the following areas:

- communication (e.g., audio/video conference systems, file transfer, email, instant messaging),
- cooperation (e.g., weblog, wiki) and
- coordination (e.g., social tagging, voting tool, application sharing tool, bulletin board) [18].

These tools can be used alone and in combination.

An example of a coordination application specialized for neighborhood work is the matching tool Zeitgeberei (www.zeitgeberei.ch), which supports neighborhood help coordinators in planning the deployment of volunteers. While Zeitgeberei enables the recording and processing of customer data, hours, appointments, and notes and the forming of tandems, it does not include any communication or cooperation components.

Social networking platforms are an example of combined application. These platforms support the establishment and maintenance of both private and business relationships on the Internet and are based on a uniform scheme of initiating, linking, and maintaining contacts [18]. However, the value of digital support depends on many individual factors, such as a technology's adaptability to the specific requirements and circumstances of a neighborhood. Neighborhood platforms have evolved as a unique form of social networking platforms. While, in principle, they function similarly to their global counterparts, they focus on a specific neighborhood or community. Examples of neighborhood platforms available in Germany include nebenan.de, nextdoor.de, crossiety.de, and Digitale Dörfer (www.digitale-doerfer.de). Although the platforms nextdoor and crossiety originate in other countries, all platforms have in common that German versions exist, and they can be used by the local coordinators without any complex configuration or programming efforts.

Regardless of the tool or platform chosen, it has to be stressed that, if a particular technology and its related services cannot be adapted to a community's needs, the expected impacts in terms of communication, cooperation, and coordination will not result [19]. Moreover, the co-creation of users in a virtual community is a key element for sustainable technology adoption [20]. Furthermore, sustainable community development involves a wide range of actions, wherein the introduction of technology is only one measure among interdependent measures.

*2.3. Implementation and Evaluation of Technology for Neighborhood Development*

The introduction of a new technology is a complex intervention [21]. Complex interventions must be well thought out and planned if they are to be successful. Progress must be continuously monitored, and measures must be readjusted if necessary. Neighborhood technology interventions follow the basic project life cycle of initiation, planning, execution, and closing. More precisely, in neighborhood development a state and problem analysis must be conducted in the beginning (phase 1), followed by goal and strategy development (phase 2) and the implementation of the technology (phase 3). Implementation success must be evaluated (back to phase 1) to redefine goals

(phase 2) and readjust measures (phase 3; see Figure 1). As neighborhoods are constantly changing, this process cycle needs to be constantly iterated.

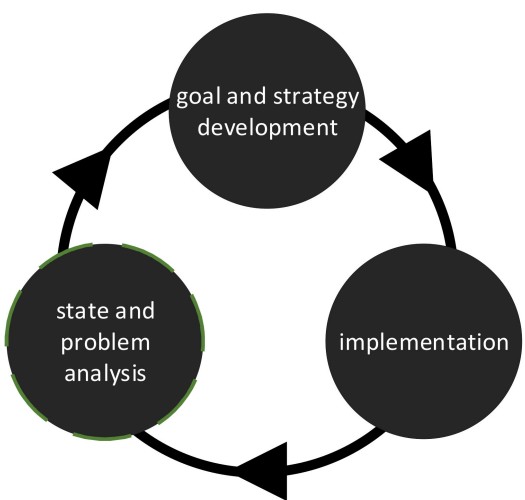

**Figure 1.** Project life cycle of a technology intervention in a neighborhood.

As stated in the introduction, it is not trivial to evaluate outcomes in the context of neighborhood development. Individual indicators are difficult to select because the multitude of aspects shaping a neighborhood make direct correlations with the introduction of technology complex. As maturity models are used to evaluate structures and processes, they might be useful in assessing possibilities for neighborhood development. A maturity model measures the ability of a project to use resources for a specific purpose [22]. In a maturity model, basic requirements are defined for processes, which are placed at different levels of maturity. A degree of maturity is assigned depending on which requirements are fulfilled. These models enable one to separately consider and analyze several dimensions of a complex process.

The body of literature regarding maturity assessment models is vast and continuously growing [22–25]; however, no maturity model is specifically tailored to technology-based neighborhood development. In the context of neighborhood development, maturity does not imply a value judgment concerning a project; rather, the level of development and capacity are evaluated in terms of their fit to a neighborhood's needs.

As neighborhood development and integrated care share many features, such as close cooperation of different local and regional health service providers, one of the many maturity models for integrated care [10,26–29], including the B3-MM [30], may be applicable to the neighborhood context.

The B3-MM categorizes the development of a regional system into 12 dimensions that contribute to integrated care [10]. The model considers each dimension of the situation within a region and assigns a measure of "maturity" (on a scale of 0–5), enabling researchers to assess the maturity level of the region or health system, including its strengths, weaknesses, and potential to foster integrated care.

## 3. Materials and Methods

To develop a maturity model for ICT-supported neighborhood development six phases were considered: (A) *Scope*, (B) *Design*, (C) *Populate*, (D) *Test*, (E) *Deploy*; and (F) *Maintain* [31]. To determine the *Scope* of the desired model, we conducted a literature review and an interview study to investigate the factors that contribute to the progress and success of ICT for community development. Six critical factors were identified for the implementation: the target group, links to existing structures, onboarding and content, the role of the caretaker, the choice of ICT, and the research context [32]. The present contribution covers the next three phases of the development process: *Design*, *Populate*, and *Test* (cf. Figure 2).

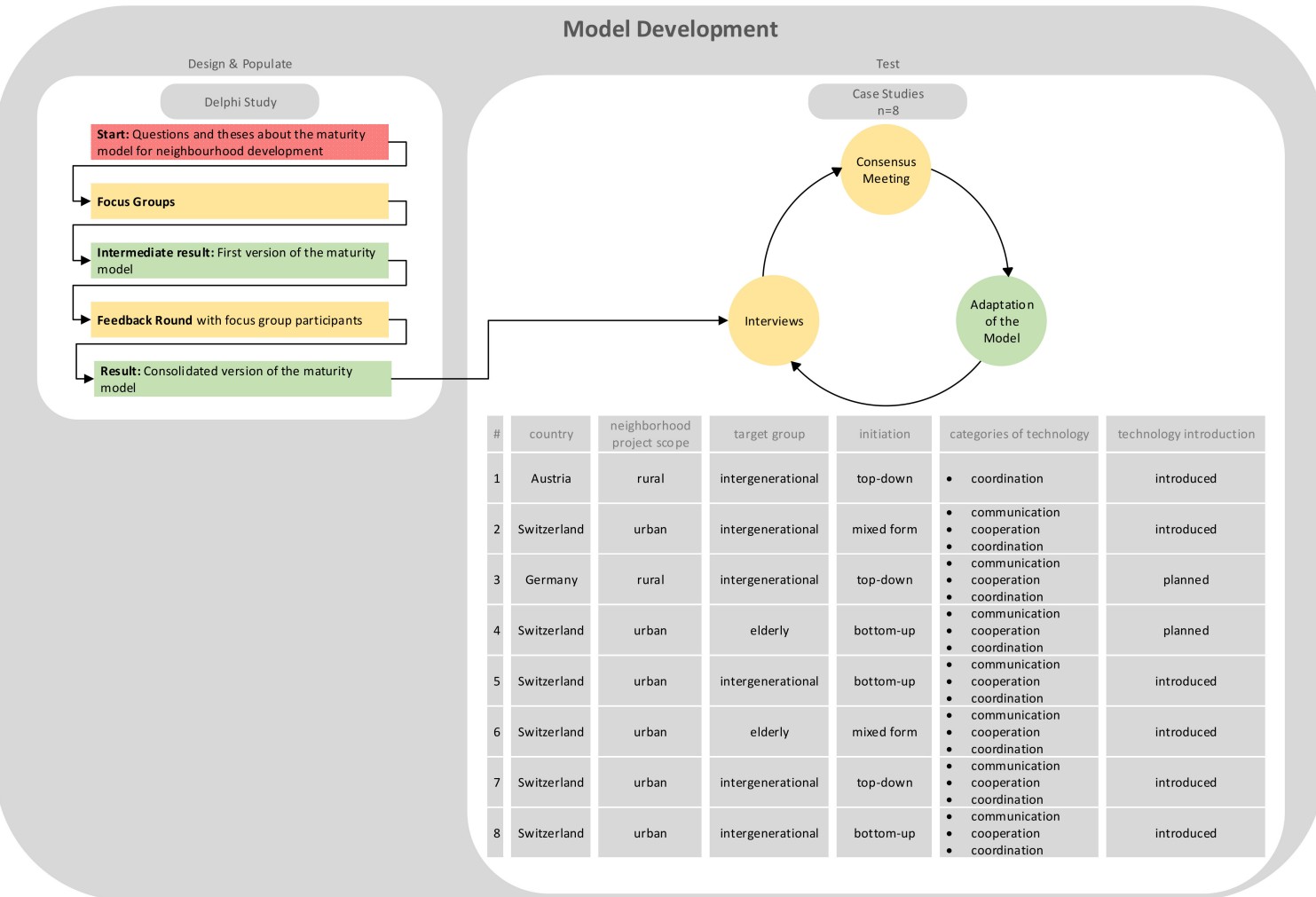

**Figure 2.** Visualization of the model development phases *Design and Populate* and *Test*; red: starting position; yellow: investigations; green: results.

## 3.1. Design/Populate Phase

The *Design* of a model forms the basis for further development and application [31]. Here, the focus is on the needs of the target group [31]. In the *Populate* phase, the content of the model is decided [31].

According to Monteiro and Maciel [33], maturity model development methods can be categorized as "conceptual," "qualitative," "quantitative," and "derivative." Having decided to base the development of the maturity model for ICT-supported neighborhood development on the B3-MM, we chose a derivative approach. In the course of the development several hurdles, such as the "linearity pitfall," the applicability of the model by the future target groups, or possibilities for knowledge transfer, have to be considered [25,34–38].

For the *Populate* phase of our maturity model, the Delphi method [31], a group communication technique with multifaceted application possibilities, was employed to examine the appropriateness of the dimensions, maturity indicators, and evaluation standards of the B3-MM, including the validity of the content for the technology-based neighborhood projects.

Two neighborhood managers, eight researchers from the social, health, information, and business sciences, and one provider of a neighborhood platform were chosen as experts for the Delphi study. The participants were all associated with the IBH "Technik im Quartier" project [39] and were proven experts in their respective areas. As part of the Delphi study, the original English version and a professionally translated German version of the B3-MM were distributed among the chosen experts. In an initial focus group workshop, all dimensions were separately discussed with regard to their transferability and relevance and any changes that would be necessary when applying them to the neighborhood context. The knowledge gained from the focus group was treated as an interim result and transferred by the authors into a first version of the maturity model. This version was then made available online to the same group of experts for additional feedback. A consolidated version of the maturity model that integrated the comments formed the basis for the next development phase.

## 3.2. Test Phase–Validation and Adaptation in Case Studies

To evaluate and further develop the model, eight case studies were conducted. The neighborhoods (one in Austria, one in Germany, and six in Switzerland) were all IBH project or associated partners and had approximately between 1000–5000 inhabitants each. In all cases, ICT solutions were used or planned to be used for neighborhood development. Figure 2 presents an overview of the case characteristics.

The case studies consisted of open interviews with one to four neighborhood managers in each neighborhood. The interviews lasted for approximately 120 min and were led by two researchers, with one moderating the discussion and the other recording it. Interviewees had the opportunity to review the maturity model prior to the interview. The first step of the interview was to determine what the participants considered a "neighborhood" and what constituted the object of investigation. Next, each dimension was discussed with regard to the current stage of the neighborhood on the one hand and the relevance of the dimension for the interviewees' neighborhood work and suggestions for possible improvements of the dimension on the other hand. At the end of each interview, results were summarized by the moderator, and participants were invited to comment. The researchers discussed the results in a subsequent consensus meeting and then fed them into a new version of the maturity model. The adaptation process was systematically documented.

## 4. Results

### 4.1. Transferability of the B3-MM to the Neighborhood Context

The Delphi study revealed that the maturity model for integrated care, the B3-MM, is partially transferable to the neighborhood context. Some dimensions largely overlap (e.g., standardization and simplification; funding; and evaluation methods), whereas others required substantial adaptations (e.g., information and eHealth services; and population approach). In contrast to integrated care,

neighborhood development is largely shaped by informal actors, and this difference was considered when defining the target group of future model users. During the translation of the model from English into German, the usage of simple language emerged as an important requirement. All 12 dimensions were regarded as important and transferable to the neighborhood context. Neither the Delphi study nor the case studies suggested a need for additional dimensions.

### 4.2. Adaptations to the Model

Figure 3 summarizes the adaptations made to the B3-MM to transform it into the **M**aturity **M**odel for digitally supported **N**eighborhood **D**evelopment (MMND). In the Delphi study, the descriptions of the dimensions and terminology were adapted to neighborhood development, whereas fine-tuning was achieved through the case studies, particularly the structure of the scale that is associated with the individual dimensions.

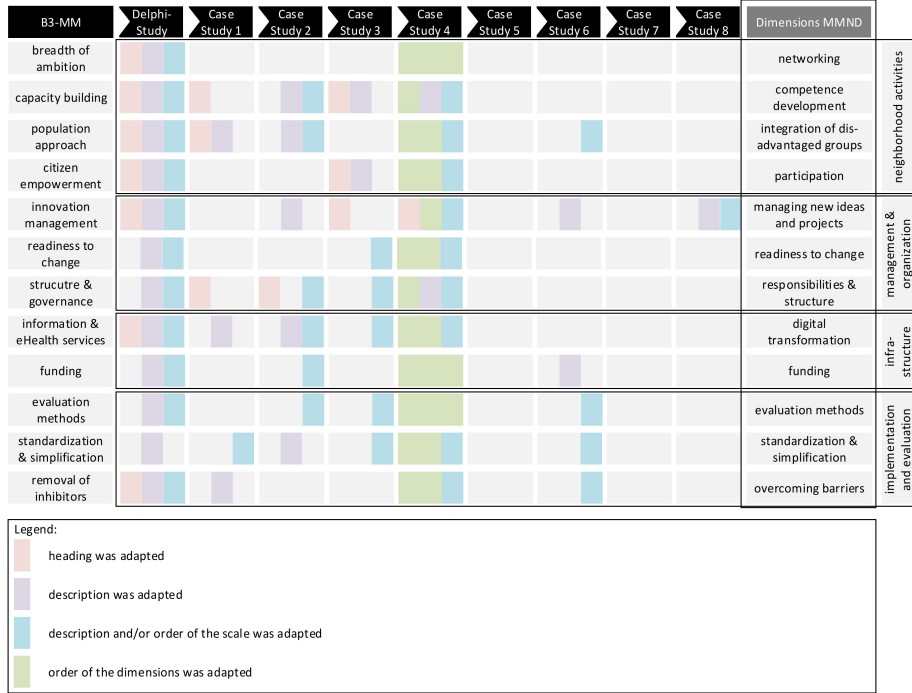

**Figure 3.** Adaptation made to the original B3-Maturity Model (B3-MM) to transform it into the final **M**aturity **M**odel for digitally supported **N**eighborhood **D**evelopment (MMND); the ordering of the dimensions corresponds to the final MMND.

The adaptations were related to both content and the level of applicability, as is apparent in the reordering and grouping of dimensions. Case study 4 clearly demonstrated that not all dimensions are equally and easily understandable. A new ordering was therefore chosen to group related dimensions and simplify the application flow.

### 4.3. Dimensions for ICT-supported Neighborhood Development

The following section presents a summary of the individual dimensions of the MMND ordered according to the categories presented in Figure 3. The whole model can be downloaded from the Supplementary Materials in English and German language (English: M1, German: M2). The model includes an introduction to the topic, instructions for application, and FAQs before the presentation of the dimensions. Each dimension includes a general description and the rating scale, as well as space for one's own comments and explanations for choosing a certain level. The model concludes with an overall evaluation in the form of a spider network diagram that visualizes the maturity of an investigated neighborhood (see Figure 4).

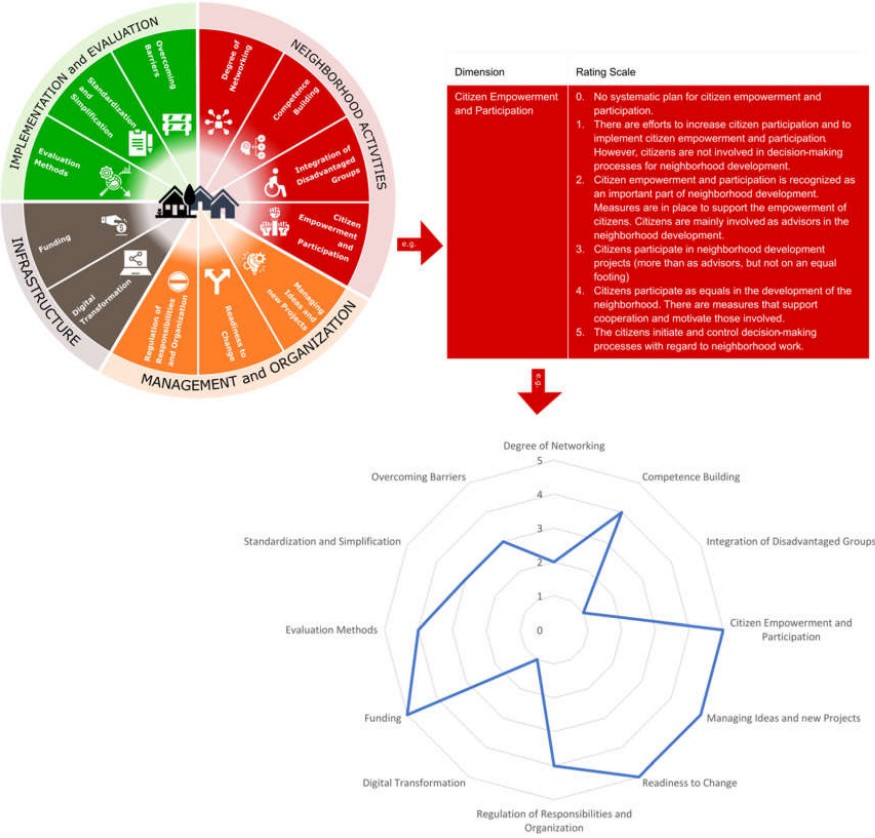

**Figure 4.** Snapshot of the MMND to be found in the Supplementary Materials: the model overview, the rating scale of an example dimension, and the spider network diagram.

### 4.3.1. Neighborhood Activities

- Degree of Networking (Degree of Integration) in the Neighborhood: A community is comprised of a wide variety of actors from different sectors including civil society, health and social services. As cooperation and exchange between actors become more formalized and more actors from different sectors participate, a broader network is formed. A long-term goal of neighborhood development is promoting systematic cooperation and exchange among all actors in a neighborhood, with the ultimate goal of achieving comprehensive community-based cooperation among the population.

- Competence Development to Foster (Volunteers') Capacity Building: In the context of neighborhood development, volunteers tend to play a crucial role. The acquisition, improvement, and maintenance of relevant skills is an important but complex task, largely due to the heterogeneity of this group. Volunteers often lack specific knowledge, including knowledge related to administrative necessities. The development of competence among citizens will increase the capacity for neighborhood development.

- Community Orientation and Integration of Disadvantaged Groups: The aim of community orientation is to improve the living conditions of all actors at local and regional levels through, with, and for local people. Often, disadvantaged people can only participate in social life to a limited extent and do not fully benefit from existing support systems. Neighborhood development projects can contribute to improving the quality of life for people in vulnerable groups and promoting their inclusion in the community. The fostering of participation and inclusion of these groups will then, as a by-product, maintain their individual health.

- Citizen Empowerment and Participation in Neighborhood Work: Neighborhood work is about the resources in the neighborhood, the strengthening of self-management and processes

of self-organization, and networking and cooperation among local institutions and actors. This dimension is concerned with enabling participation of citizens as co-designers of change processes. To address this dimension, the population must be provided with easy-to-use tools that promote their involvement in neighborhood development, such as technical solutions that allow people to express their opinions.

### 4.3.2. Management and Organization

- Managing Ideas and new Projects: Many of the best ideas are likely to come from committed neighborhood residents or professionals who understand where improvements can be made to existing processes. These innovations need to be recognized, assessed and, where possible, scaled up to provide benefits across the neighborhood. The focus of this dimension is not on the technological or material but on social innovations (e.g., using ICT for neighborhood development).
- Readiness to Change to Community-based Development: Change in existing systems is often accompanied by the creation of new roles, processes and working practices. Such change requires a broad-based motivation for change and a strategy and vision of how neighborhood cooperation should be shaped in the future.
- Regulation of Responsibilities and Organizational Structure: The structuring of neighborhood work and higher-level governance is not a guarantee of success. In the context of neighborhood development, control at the regional or local level is preferable. Structure is seen as a service provided by the municipality to the citizens. This service can be used on a voluntary basis for project development but is not mandatory to use.

### 4.3.3. Infrastructure

- Digital Transformation: A lively neighborhood relies on communication, exchange and community. Transparency and communication between citizens and local professional actors and institutions are an important basis for effective neighborhood work. In contrast to integrated care, digital information and communication services are used to support community work and interaction in a neighborhood. These services should support the efficient cooperation of community actors and enable citizens and multipliers to interact and participate in the neighborhood development. Digital services are ideally based on existing offerings and structures. Digital interaction possibilities should also be based on and extend existing networks.
- Funding: Successful and sustainable community development requires an initial investment at both organizational and technical levels, as well as continued financial support until new structures and services are fully operational. Ensuring the financing of initial and running costs is therefore an essential measure.

### 4.3.4. Implementation and Evaluation

- Evaluation Methods: The evaluation of interventions for digitally supported neighborhood development is often a prerequisite for funding and official recognition. Evaluation is also a tool for standardizing procedures and thus facilitates the exchange and transfer of best practices.
- Standardization and Simplification: Standardization of procedures and implementation strategies for ICT can simplify collaboration among all involved actors. Furthermore, the exchange of guidelines and best practices between projects and neighborhoods could be desirable.
- Overcoming Barriers: Good technologies alone do not guarantee successful neighborhood development. User-friendly and self-explanatory technology may be subject to other barriers, including a lack of political support or insufficient marketing, which may prevent the spread of the technology. The "removal of inhibitors" may not always be possible in neighborhood development; therefore, this dimension must focus on how to manage these barriers.

## 5. Discussion

This paper illustrated the development of an instrument that supports the exchange of best practices and promotes the implementation and scaling up of community-based initiatives with ICT support. Our results revealed that the B3-MM is partially transferable to the neighborhood context. Different application scenarios can thus be discussed for the newly developed model.

### 5.1. Application of the Model

The case studies and the literature (e.g., [22,25,33,38]) provide several examples of application scenarios for the MMND:

- Self-Assessment The MMND can be used as a self-assessment tool with which a neighborhood developer or a team of neighborhood developers can evaluate the current state of a project. The results of the self-assessment could also be communicated to existing or potential funders if required or desired.
- Tool for consensus building The MMND could be the basis for a consensus process between diverging neighborhood stakeholders. In this case, each stakeholder or stakeholder group would conduct a self-assessment prior to a consensus-building workshop. In the workshop, the self-assessment results would be compared and discussed to reach consensus about what strategy to adopt and how to proceed in the future.
- Planning Tool The model can also serve as a basis for comprehensive project planning. In this case, the tool could be used for orientation at the beginning of a project. A planning workshop should consist of three steps: First, the status quo of the neighborhood should be assessed for each dimension. Second, a maturity target for each dimension should be determined. Finally, the necessary steps to achieve the target must be defined.
- Benchmarking/Exchange of Best Practices On the basis of prior self-assessments, it could be desirable to compare and exchange experiences between neighborhoods. The eight case studies demonstrate that accompanying materials and procedures would need to be standardized and made easily accessible to achieve this exchange of ideas.
- Monitoring The maturity model could be used to monitor the achievement of milestones and project progress at regular intervals. The majority of practitioners felt that, if the model was used repeatedly over an extended period of time, progress would be made visible.

Even though maturation is defined as "development toward the better" [22], "better" can be achieved in different ways. The case studies showed that the MMND can be applied at different stages of a project with various aims (see Figure 5) and that the focus is not necessarily on reaching the next hierarchical level, but on initiating discourse and discussion of the topic.

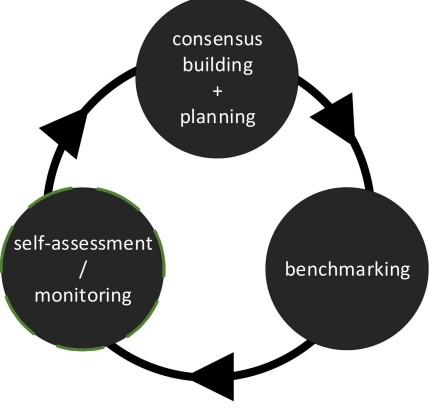

**Figure 5.** Applicability of the model in the project life cycle adapted from Figure 1.

The repeated application of the maturity model is particularly valuable to highlight the interdependencies between the dimensions and enable continuous adjustment. As an example, the dimensions "Citizen Empowerment and Participation in Neighborhood Work" and "Digital Transformation" interact in every phase: Empowerment and participation are necessary, in the "state and problem analysis" phase to determine the problems in a neighborhood that are most urgent for citizens. Technology could support the gathering of problems and feedback (e.g., in the form of online surveys). However, the target group must be empowered to use this technology. If all target groups are not equally empowered, the analysis risks being biased. The same is true for the "goal and strategy development" phase, wherein the objectives are defined and the technology that fits the needs of the target group is chosen. The citizens, as the target group, should be empowered to participate in these phases, as otherwise they may not accept the chosen technology or support its implementation. Applying the maturity model in each project phase can prevent biased analyses and enable taking on (technology) measures to enhance participation.

### 5.2. Target Users of the Model

According to our results, neighborhood managers were the target users of the model, and the model can be applied by a single person as well as by a group of people. The instrument was helpful in promoting exchange within the neighborhood development team. Participants said that the maturity model helped them understand the current situation in their neighborhood. However, when the model is used by a group, the group's size is an important factor. For example, Wheelan [35] has argued that smaller group sizes are more manageable and productive, which was confirmed by this study's positive experience of using the model as a self-assessment tool in small groups with up to four participants (plus two researchers). For this application scenario, the primary target group consisted of decision-makers at the neighborhood management level who have the necessary knowledge of a neighborhood's specificities and decision-making structures.

To enable later comparisons of maturity levels, it is advantageous to carry out the assessment with the same people in each instance. As this may not always be possible, it is crucial that the subject of the study and the choice of levels be described in a detailed report.

### 5.3. Defining the Object of Investigation and Maturity Level

Digitally supported neighborhood development is a complex endeavor. The case studies have shown that evaluations from the perspective of a single project or initiative can be difficult. The success of an individual project is strongly dependent on the conditions of a neighborhood. Furthermore, certain processes are not decided at the neighborhood level. A differentiation between the perspectives of "city as a whole" and "neighborhood" was necessary at times, and the dimensions of the model were discussed from both perspectives. Some participants concluded that the instrument's operational capability is greater at the town level than at the project or neighborhood level.

It is necessary to investigate and document neighborhood conditions to compare maturity levels. The eight case studies conducted in three countries with differing political, cultural and financial frameworks illustrate the heterogeneity of neighborhood development. Although the municipalities were involved in each project in some capacity, governance was initiated and/or controlled by differing entities, such as a nursing service provider, civil society representatives or the municipality. The technology implementation procedures also differed between cases.

For stage-based models the hierarchy of the scales generally presents an issue. To avoid the "linearity pitfall" [25], we combined different possible sequences for different stages. In some of the case studies, the granularity of the maturity levels presented an additional issue. In the Swiss case study 2, the maturity levels were considered too restrictive, whereas stricter and more granular monitoring was desired in the German case study 3. In part because of these differences, several adaptation rounds were required to define the maturity scales. However, the maturity scales may still be unsuitable in some cases.

This study demonstrates that maturity assessments can be carried out at differing stages of a project and can be applied to different study objectives. In case 1, for example, technology was introduced several years ago, whereas digitization has not yet played a role in all other cases until recently.

### 5.4. An Excurs: The Impact of the Coronavirus

Currently, providers of community collaboration technology are reporting an exponential increase in interest in their products [40,41]. Simultaneously, municipalities have been announcing their up-coming introduction [42] or their successful implementation [43], especially since the Coronavirus pandemic. While in some instances the community platforms have enjoyed great acceptance, others have been less successful and have been terminated after only a short period of time. While this shows that many decision-makers have recognized the potential and value technical networking can have, it also highlights the unknowns when it comes to the implementation of technology. While the Coronavirus pandemic may have boosted the introduction of technology in some cases, it has not changed the fundamental complexity and challenges related to the actual use and acceptance of technology. Based on the insights that we have gained through our case studies, we can conclude that, if neighborhoods engage in strategy development at an early stage, account for their specific conditions and structures, and continuously monitor the progress of their projects using the maturity model, the chances of successfully establishing a thriving community-based technology measure will increase.

### 5.5. Limitations and Future Work

The B3-MM model, which served as the starting point of the MMND, is a widely tested and used evaluation instrument [38]. While we observed a certain saturation of necessary changes after the sixth case study, the fact that the MMND was tested with only eight cases is a limitation of the present work. The knowledge gained in this study should therefore be considered a first step toward validating the model. As Uijen et al. [34] state, this limitation does not imply that the model is of low quality but rather that there is need for further quantitative studies. To foster such studies, the English and German versions of the MMND were added as Supplementary Materials. The English version was generated after the development of the German version. For the case studies, only the German version was used.

With regard to the development process, and the Delphi method in particular, the objection could be raised that a wider variety of experts could have been consulted. Although the Delphi technique is a widely applied method, there are no universally agreed upon criteria for the selection and number of experts or the number of rounds [10]. The chosen process should therefore be judged according to the quality of the experts and not their representativeness for statistical purposes.

The lack of ready-to-use documentation [22] must also be acknowledged. Throughout this study, technological and/or organizational support were mentioned as desirable additions. This support could take the form of a moderator, as in the case studies, or could be achieved through additional materials and guidelines. Given the diversity of neighborhood development projects, a comparison of the results is challenging without a standardized procedure for describing and documenting individual cases. There is a need to create handbooks and accompanying materials to reduce the abstractness of the model for practitioners. In the eight case studies, the MMND was applied and discussed in a single meeting. It is conceivable that the assessment could be conducted in several discussion rounds with fewer dimensions addressed in each. However, we observed that some dimensions were perceived as more challenging than others. It is therefore recommended that project teams conduct an overview of all dimensions and do not end meetings with those that may be most challenging for a particular project. The presented order of the dimensions is a suggestion and should be adapted to the needs of each individual neighborhood. Current research on maturity models is concerned with how to apply these theoretical models in a practical context [36] to help stakeholders and encourage a continuous learning cycle (e.g., Plan-Do-Act-Check [37]). To use maturity models in a practical context, tools such as checklists with point systems, recommendations for how to move from one maturity

level to the next, and information about when to promote change could be helpful. These tools have yet to be developed for the MMND. Finally, it is important to acknowledge the fact that achievement of the highest possible maturity level is not necessarily useful for every neighborhood project.

## 6. Conclusions

Good technologies alone are not enough to successfully develop a neighborhood and encourage cooperation. To achieve a successful transformation to smart and socially sustainable neighborhood development, it is necessary to understand which factors contribute to this goal. This article presented a maturity model for digitally supported neighborhood development.

Among other things, the eight case studies demonstrated that the MMND can be applied at different stages of a project with various aims, such as supporting the initial planning process, discovering possible gaps or barriers, identifying areas for improvement, or encouraging discussion within the project team. To ensure the most effective use of the maturity model in the context of technology-supported neighborhood development and to foster the exchange of best practices and experience between projects, accompanying materials and services must be easily accessible. As with other maturity model projects [38], the organization of "twinning and coaching" could conceivably promote an ongoing learning process. Through this process, it may be possible to compare projects in different settings and countries to identify their complementary strengths and weaknesses, as well as possible gaps.

Like most projects, neighborhood development is often under external pressure related to financial factors. Funding is often only provided for innovative projects. However, our experience is that digitization projects are more successful if they are based on established structures. Furthermore, the providers of funding or financial support generally require project evaluation and monitoring. As a standardized instrument, the MMND could on the one hand provide a simple and cost-effective method for assessing and monitoring the progress of projects and on the other hand help prepare the ground so that the digitization project can be based on sustainable physical structures.

**Supplementary Materials:** The following are available online at http://www.mdpi.com/2071-1050/12/22/9319/s1, Model M1: Maturity Model for digital Neighborhood Development, Model M2: Reifegradmodel für digitale Quartiersentwicklung.

**Author Contributions:** The authors are an interdisciplinary team with a wide range of competences ranging from social work, health promotion and literacy, information and computer sciences, to economics. They have extensive experience of neighborhood development which has been gained in several large-and small-scale studies. All authors have contributed to this manuscript. Conceptualization, M.R., A.H. and E.M.; Formal analysis, M.R., A.H. and E.M.; Funding acquisition, C.K.; Investigation, M.R., A.H. and E.M.; Methodology, E.M., F.T. and C.K.; Project administration, M.R.; Supervision, F.T. and C.K.; Visualization, M.R.; Writing–original draft, M.R.; Writing–review and editing, M.R., A.H., E.M., F.T. and C.K. All authors have read and agreed to the published version of the manuscript.

**Funding:** The MMND was developed in the context of the IBH project "Technik im Quartier" (V-2.0101/ABH067). The project is funded by the European Union and the European Regional Development Fund in the Interreg V-Programm "Alpenrhein-Bodensee-Hochrhein". The jobs of Madeleine Renyi and Anna Hegedüs were partly financed by this program.

**Acknowledgments:** We would like to thank all IBH neighborhood actors, who supported us throughout the study. We would also like to thank all experts who contributed to this study by sharing their knowledge.

**Conflicts of Interest:** The authors declare no conflict of interest. The funders had no role in the design of the study; in the collection, analyses, or interpretation of data; in the writing of the manuscript, or in the decision to publish the results.

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
