# Peer review of "Toward Sustainable ICT-Supported Neighborhood Development—A Maturity Model"

_sustainability, doi:10.3390/su12229319_

Round 1

Reviewer 1 Report

In today's society, the development of neighborhoods and neighborhood groups is becoming increasingly important. New technologies have a crucial role to play in this sustainable development.

The document addresses an important and interesting topic of analysis of the impact of new technological advances in the field of neighborhood.

This is a very well structured and comprehensive paper. It is very relevant and timely given the chosen topic.

The summary is correct because it helps the reader understand the research. It describes the methodology used with the application of the Delphi method and the number of cases analyzed.
The application of the study to the effects of the coronovirus pandemic is introduced but I have missed some other reference in the paper.
It is very important that the summary reflects the research to be done and does not create expectations that are not satisfied afterwards.

The Introduction includes a basic literary review. In fact, in the Discussion, new bibliographical references that had not been included in the introduction are added.

The use of the Delphi method in the methodology and the sample selected in several countries is very interesting.

The results are limited to a description of the MMND. It was expected that the model would be applied to the life of the neighborhoods, especially now with the COVID-19.

Therefore, the Discussion and Conclusions expected more input from the authors on this new society and the importance of neighborhoods in it, and not just a description of the MMND model. I believe that the manuscript will contribute significantly to the readers.

The selected research is very interesting and can contribute a lot to the life of the neighborhoods. In my opinion, this paper can be very interesting if the authors get more involved in the results and conclusions and analyse the contribution that new technologies can affect the people living in the neighbourhoods.

Reviewer 2 Report

I read your article with interest and confess that, while I found your model intelligible, if offered at a very high level of aggregation, I was very uncertain of exactly how you think it could or should be employed. This was so across virtually every dimension of your frame. You suggest, for example, that you will seek digital transformation, but I was never clear what might be necessary to secure that possibility, who would work to ensure it or how it would relate to other elements of your model or residents. Obviously, labeling it "transformational" set a very high bar in the first instance. To cite another example, you appropriately suggest that citizen empowerment and participation are vital to community development and I suspect that the IBH project, at least,is government driven and has specific goals for such efforts, but I do not know what those are from your article nor how participation might be important to them nor what strategies might be in use to secure citizen engagement and concerning what. And the "what" obviously matters. So, as your reader, I find myself being asked to make sense of a very broadly framed normative model whose elements are all quite general and vague and to do so without any real context of what sort of neighborhood development is envisaged (what exactly do you mean by a caring community, for example?) and why or who or what players might be leading this very generally framed initiative. This conceptual muddle was exacerbated for me by the fact that your abstract is written largely in passive voice so I could not glean who actually would be undertaking to establish and maintain the "community based technologies" to which you refer. All of this said, as an overarching proposition, I found what you share in figure 3 regarding relevant factors is salient, but it is so in only the most generalized of senses. Any number of major initiatives would be necessary in situ to address any one of the categories you identify. So, do you see this model simply as an organizing heuristic at the proverbial 50,000 foot level? If so, who do you think would employ it and how and how, precisely, would information and communication technology be employed to pursue those purposes?

Round 2

Reviewer 1 Report

The new version revised by the authors corrects the errors detected previously in a satisfactory way.

The document deals with an important and interesting subject of analysis of the impact of new technological advances in the area of the neighbourhood.

The new summary currently reflects the study being developed.

The division between introduction and theoretical framework has placed the problem more clearly. Improving the document substantially

In spite of the expanded background the literature review has hardly been improved.
More references that situate the research are missing.

It describes the methodology used with the application of the Delphi method and the number of cases analyzed.

The use of the Delphi method in the methodology and the sample selected in several countries is very interesting.

The results and the discussion have been improved in the new version

In my opinion the changes made to the work substantially improve the research. The selected research is very interesting and can contribute a lot to the life of the neighborhoods.

Reviewer 2 Report

I have reviewed your revision with interest. I continue to find it quite abstract but I do see the value of the model you present as a heuristic, subject to the substantial caveat that realizing any number of its elements will be subject to realization of a host of conditions and only over (likely) a considerable time. This caveat includes, of course, any and all use of ICT as an element of a development strategy, especially within poor or otherwise vulnerable communities. I note too, if parenthetically, that it strikes me that "caring communities" depend mightily on pre-existing social ties, especially across difference, which are being tested strongly in Europe and the United States and in many other nations around the world as I write. AN ICT platform cannot alone create those ties or the social solidarity on which they are predicated.